# Study on Low-Temperature Cracking Resistance of Carbon Fibre Geogrid Reinforced Asphalt Mixtures Based on Statistical Methods

**DOI:** 10.3390/polym17040461

**Published:** 2025-02-10

**Authors:** Yifan Huang, Zhiqiang Wang, Guangqing Yang

**Affiliations:** 1Key Laboratory of Roads and Railway Engineering Safety Control (Shijiazhuang Tiedao University), Ministry of Education, Shijiazhuang 050043, China; huangyf@stdu.edu.cn (Y.H.); yanggq@stdu.edu.cn (G.Y.); 2School of Traffic and Transportation, Shijiazhuang Tiedao University, Shijiazhuang 050043, China

**Keywords:** carbon fibre geogrid, surface combined body, low-temperature bending damage test, linear fitting difference method, two-way analysis of variance

## Abstract

In order to investigate the effects of surface combined body (SCB) type and geosynthetic type on the low-temperature cracking resistance of reinforced asphalt mixtures, low-temperature bending damage tests were conducted on both unreinforced and reinforced double-layer beam specimens, respectively. At the same time, the load–deflection curve during loading was corrected using the linear fitting difference method to determine the mid-span deflection. Then, the low-temperature cracking resistance of the reinforced asphalt mixtures was comparatively analyzed by calculating the maximum flexural tensile strain (*ɛ*_B_). Finally, the extent to which the geosynthetic type and the SCB type affect the low-temperature cracking resistance of the reinforced asphalt mixtures was investigated by means of a two-way analysis of variance (ANOVA). The results showed that the greater the tensile strength of the geosynthetics, the greater the mid-span deflection and *ɛ*_B_ of the reinforced double-layer beam specimens. The order is carbon fibre geogrid (CCF) > glass/carbon fibre composite qualified geogrid (GCF) > fibreglass–polyester paving mat (FPM) > unreinforced (UN). In the case of reinforcement, the *ɛ*_B_ of the AC-13/AC-20 combination is lower than that of the AC-20/AC-25 combination, with a significant difference, especially in the case of geogrid reinforcement. Analysis by a two-way ANOVA shows that the order of influence on *ɛ*_B_ ranks as geosynthetic type > SCB type. This study provides a scientific basis for the rational selection of carbon fibre geogrid–reinforced asphalt pavement structures.

## 1. Introduction

With the rapid growth of the national economy, the construction of highways has flourished. By the end of 2023, the total mileage of highways in China exceeded 183,600 km. Highways are playing an increasingly important role in promoting economic development in areas along their routes, generating enormous social benefits. Due to asphalt pavement’s advantages of high mechanical strength, good flatness, jointless construction, good wear resistance, and a short construction period, the vast majority of highway pavements in China have been selected in the form of asphalt pavement [1,2,3]. With the continuous increase in traffic volume, axle load, and overloading, coupled with the influence of the external natural environment, asphalt pavement frequently suffers from diseases such as cracks under the combined effect of temperature effects and traffic loads.

Cracking remains one of the major diseases of asphalt pavements. Cracking accelerates the deterioration of the asphalt pavement structure and greatly shortens the service life of the pavement [4,5,6]. The cracking of asphalt pavement mainly results from low-temperature cracking [6,7,8]. Many domestic and foreign scholars have carried out research on the low-temperature cracking resistance performance of asphalt mixtures to address the problem of asphalt pavement cracking [3,9,10] but still have not been able to fundamentally solve the cracking problem. Presently, geosynthetic-reinforced structures, which have remarkable economic advantages and outstanding technical performance [11,12,13], have been extensively used in transport infrastructure, such as railways and highways. Therefore, numerous scholars have conducted a series of studies on the performance of geosynthetic-reinforced asphalt pavement.

Khodaii et al. [14] studied the effects of the type of existing surface layer and the geogrid position on the cracking resistance of asphalt overlay through Type I fracture tests. It was shown that reinforcement significantly reduced the rate of crack expansion. Wang et al. [15] investigated the cracking resistance of interlayer materials by using a bending tensile test method, and the cracking resistance effect was as follows: glass fibre geogrid > geotextile > uninterlayer material. Nejad et al. [16] researched the influence of the modulus of geosynthetics on the number of loading cycles needed for crack emergence and extension in reinforced asphalt mixtures through cyclic loading tests, and revealed that the modulus of the geosynthetic influenced the number of loading cycles before failure and the rate of crack extension. Gonzalez-Torre et al. [17] investigated the influence of geosynthetic type and the secant modulus on cracking resistance performance with different traffic loads and temperatures. The results demonstrated that the addition of geosynthetics significantly impeded the propagation of cracks and improved cracking resistance performance. Nejad et al. [18] studied the effect of geosynthetic type on the expansion of reflective cracks in asphalt overlays and found that glass geogrids effectively improved the cracking resistance of the asphalt overlay. Saride et al. [19] discovered that the inclusion of geosynthetics enhanced the cracking resistance of asphalt overlays, but the problem of delamination occurred. Lee et al. [20] researched the effect of the fibre geogrid type on the long-term performance of asphalt overlay pavements utilizing field tests, finite element analyses, and laboratory tests. The results indicated that the placement of the carbon fibre geogrid could reduce the cracking rate and improve long-term performance. Ling et al. [21] found a geotextile with better resistance to reflective cracking than an interlayer structure. Solatiyan et al. [22] studied the effect of geotextiles on the mechanical properties of asphalt overlays through a three-point bending test. The results indicated that geotextile reinforcement was beneficial in increasing the mechanical properties. In addition, several studies have shown that geogrids, as a reinforcement material for asphalt mixtures, have a significant effect on the propagation process of cracks [23,24,25].

In summary, the reinforcing effects of geogrids are mainly composed of two parts: lateral constraint and tensiled membrane effects [14,26]. Lateral restraint is mainly provided by the surface friction between the geogrid and the asphalt mixture, the surface cohesion, and the bite force between the geogrid and the asphalt mixture. The bite force includes the end-bearing capacity between the geogrid ribs and the asphalt mixture, and the surface friction force between the asphalt mixture inside the grid holes and the upper asphalt mixture outside the holes. The bite force between the geogrid and asphalt mixture is different from that of geotextile-reinforced materials, but also embodies the superiority of the geogrid. Under the action of the load, the geogrid undergoes a certain deformation, generating an ensiled membrane effect, which provides vertical support and reduces vertical deformation.

There are other factors that contribute to the low-temperature cracking of asphalt pavement structures, including asphalt type, asphalt composition, porosity, fibre quality, aging, and others [27,28,29,30,31]. Teltayev et al. [28] investigated the effect of different polymer modifications on the low-temperature cracking resistance of asphalt binders and their mixtures and found that polymer-modified asphalt mixtures had a stronger tensile strength at low temperatures. Budzński Bartosz et al. [31] researched the effect of porosity and asphalt type on the low-temperature performance of asphalt, and proposed that asphalt type was the key factor, while porosity was secondary. Lin et al. [32] found that fibres that were subjected to short- and long-term aging improved the cracking resistance of their modified asphalt mixes. Ren et al. [33] examined the effect of the degree of aging of asphalt on its relaxation behaviour and suggested that the long-term aging of asphalt enhanced its relaxation time.

The above literature mainly concentrates on the cracking resistance performance of geosynthetic-reinforced asphalt pavements, particularly glass fibre geogrid–reinforced asphalt pavements. However, there are few studies on the cracking resistance performance of carbon fibre geogrid–reinforced asphalt pavements. Statistically based studies on the cracking resistance of reinforced asphalt pavements are rarely conducted. Carbon fibre is mainly used in certain high-end applications, such as aerospace, automotive, and sports, mainly due to the high cost of carbon fibre production. The main reason for its high cost is the high cost of the precursor yarn, accounting for well over 50% of the total cost [34,35]. Giorgio et al. [36] analyzed the force versus displacement maps of parallel, pantotactic, and isostatic textures; they found a difference of several orders of magnitude in the displacement force maps of the considered textures and justified their behavioural characteristics. dell’Isola et al. [37] analyzed the current state of research on metamaterials and suggested that the focus should be on strongly interacting inextensible or nearly inextensible fibres. Reducing the cost of carbon fibres is a current trend. The use of textile-grade polyacrylonitrile (PAN) fibres (Tex-PAN) as a low-cost precursor is an effective approach [35,38]. Recently, some scholars have succeeded in the preparation of Tex-PAN-based carbon fibres [35,38,39]. Nowadays, the cost of Tex-PAN is only 2~3 USD/kg, while the price of the carbon fibre precursor PAN primary filament is 7~8 USD/kg. Compared with the dedicated primary filament for carbon fibre, the manufacturing cost of Tex-PAN primary filament can be reduced by about 65%. Under the same conditions, it can reduce the total cost of production of PAN-based carbon fibres by about 35% [40]. Carbon fibre geogrids are a new type of warp-knitted geogrid [6] with less susceptibility to temperature that has excellent mechanical properties and good engineering properties. Nevertheless, the research on carbon fibre geogrids to prevent and control the problem of cracking in asphalt pavement is in its initial stages in China.

From the above analysis, the purpose of this paper aims to provide in-depth insight into the low-temperature cracking resistance of carbon fibre geogrid–reinforced asphalt pavements through statistical methods. In addition, this paper presents a method for determining the mid-span deflection in low-temperature bending damage tests. Therefore, this paper mainly investigates the effects of geosynthetic type and surface combined body (SCB) type [6] on the low-temperature cracking resistance of reinforced asphalt pavements. Based on low-temperature bending damage tests, firstly, the mid-span deflection was determined by the linear fitting difference method to obtain low-temperature cracking resistance parameters. Secondly, the effects of the geosynthetic type and SCB type on the maximum flexural tensile strains (ɛ_B_) are discussed. Finally, the extent to which the geosynthetic type and the SCB type influenced the low-temperature cracking resistance of the SCB was determined by two-way ANOVAs with interactions. This study contributes to better comprehension of the low-temperature cracking resistance of reinforced asphalt pavement structures and provides an important theoretical basis for the rational selection of new asphalt pavement structural forms.

## 2. Materials and Methods

### 2.1. Materials

#### 2.1.1. Geosynthetics

The geosynthetics used as reinforcements for this study are fibreglass–polyester paving mats (FPMs), carbon fibre geogrids (CCFs), and glass/carbon fibre composite-qualified geogrids (GCFs), as shown in Figure 1. FPMs are composed of glass and polyester fibres, a new glass fibre composite and anti-cracking material, as shown in Figure 1a. Its properties are shown in Table 1. CCFs are made up of carbon fibre rovings, as shown in Figure 1b. GCFs are composed of carbon fibre rovings and glass fibre yarns, with transverse ribs made of carbon fibre rovings and longitudinal ribs made of glass fibre yarns, as shown in Figure 1c. The technical specifications of the geogrids are shown in Table 2.

#### 2.1.2. Asphalt

The asphalt used as binder material in this study was styrene–butadiene–styrene block copolymer (SBS)-modified asphalt. According to the demands of the Chinese specification JTG E20–2011 [41], various indicators of SBS modified asphalt were tested, as illustrated in Table 3. All indicators met the demands of the Chinese specification JTG F40–2004 [42]. PCR cationic emulsified asphalt was selected as the adhesive layer, with a dosage of 0.4 L/m^2^, as indicated in Table 4.

#### 2.1.3. Surface Combined Body (SCB)

The asphalt mixtures used in this study were dense-graded asphalt concrete mixtures (AC)—AC-13, AC-20, and AC-25—and their aggregate gradation curves are shown in Figure 2. The asphalt aggregate ratios for AC-13, AC-20, and AC-25 were obtained from the Marshall test and were 4.8%, 4.3%, and 3.7%, respectively. Basalt and limestone were used as aggregates for AC-13. On the basis of actual Chinese asphalt pavement [43,44,45], the aggregate with a nominal particle size of 10–15 mm was basalt, as was the aggregate with a nominal grain size of 5–10 mm. Limestone was used as the aggregate for all the remaining asphalt mixtures [46]. The SCB was the asphalt pavement structure that consisted of the upper and lower layers of the asphalt mixture and the tack coat [46]. The SCBs used in this study are AC-13/AC-20 and AC-20/AC-25.

### 2.2. Specimen Preparation

Based on a series of low-temperature bending damage tests, the effects of the geosynthetic type and SCB type on low-temperature cracking resistance were comparatively analyzed. The unreinforced SCB (UN) was used as the control group and the reinforced SCB as the test group. Three geosynthetic types (i.e., PFM, CCF, and GCF) were chosen. Two SCB types, AC-13/AC-20 and AC-20/AC-25, were selected. The above SCBs were designed for different interface locations of actual asphalt pavement structures to facilitate construction. The low-temperature cracking resistance of reinforced asphalt pavements with different geosynthetics at different interface locations was simulated.

The procedure for preparing the double-layer reinforced beam specimens used in this study was as follows: (a) Mixing and rolling to form the lower layer. The asphalt mixture was mixed according to the requirements and poured into the rutting test mould with dimensions of 300 mm × 300 mm × 50 mm, and the lower layer was rolled and shaped in accordance with the demands of the Chinese specification JTG E20–2011 [41]. (b) Laying geosynthetics. The prepared lower layer was loaded into the rutting test mould with dimensions of 300 mm × 300 mm × 100 mm, and uniformly coated with PCR cationic emulsified asphalt and laid with geosynthetics. (c) Mixing and rolling to form the upper layer. The upper layer was obtained in a similar way to create a double-layered, reinforced rutting plate. (d) Plate cutting. The specimen preparation process is depicted in Figure 3. The preparation process of the specimen was described in detail in the literature [6]. To compare the unreinforced SCB to the test SCB, this experiment also prepared unreinforced double-layer beam specimens (UN). The dimensions of the double-layer reinforced beam specimen were 250 mm × 47 mm × 50 mm, as indicated in Figure 4.

### 2.3. Low-Temperature Bending Damage Test

The apparatus used for the low-temperature bending damage test were the WDW-1020 micro-controlled electronic universal testing machine and the WD-402 high- and low-temperature test chamber. In this test, prismatic specimens were used. The displacement loading mode was selected. The loading rate was 50 mm·min^−1^. The test temperature was −10 °C. The termination condition of the test was if the crack expanded to 80% of the height of the beam specimen. This test is shown in Figure 5. The test parameter was the maximum flexural tensile strain, which was calculated as follows:(1)$$\varepsilon_{B}=\frac{6hd}{L^{2}}$$
where *ɛ_B_* is the maximum flexural tensile strain of the specimen when damaged; *d* is the mid-span deflection of the specimen when damaged, mm; *h* is the height of the cross-interrupt interview piece, mm; and *L* is the span of the specimen, mm.

## 3. Results

### 3.1. Determination of the Mid-Span Deflection

In order to eliminate the effects of compaction and bearing contact at the beginning of loading, it is necessary to correct the starting part of the load–deflection curve. The straight line segment of the load–deflection curve in Figure 6 should be extended according to the method shown in the diagram, and intersect with the horizontal axis as the starting point of the curve in order to measure the mid-span deflection (*d*). In this study, the linear fitting difference method was used to determine the straight line segments of the load-displacement curve to obtain *d*, as described above.

The linear fitting difference method used Origin software (Origin95_64) to obtain different straight line segments of the load–displacement curve through linear fitting. The horizontal coordinates of the peak values were substituted into the different fitted linear equations to obtain the respective predicted value loads, which were then differed from the peak loads. The fitted linear equation with the smallest difference was selected as the equation of the required linear segment to obtain the mid-span deflection (*d*).

Taking the GCF-reinforced beam specimen with the AC-20/AC-25 combination as an example, *d* was calculated with the linear fitting difference method. The specific process was as follows:(1)Take points on an approximate straight line, with the peak point as the first point, followed by the second point, the third point, etc. The GCF_1_ group performed linear fitting on three points, four points, five points, six points, seven points, eight points, nine points, and ten points, respectively. The GCF_2_ performed linear fitting on three points, four points, five points, six points, seven points, eight points, nine points, and ten points, respectively. The GCF_3_ group performed linear fitting on three points, four points, five points, six points, seven points, eight points, nine points, and ten points, respectively. The respective linear segment equations were obtained, as shown in Table 5.(2)The predicted value of peak load was obtained by substituting the horizontal coordinates of the peak point into the obtained linear segment equation. Differences were made with the peak load value to obtain the difference and percentage. The results are shown in Table 6.(3)The line equation corresponding to the minimum difference or percentage obtained through comparison was the obtained line equation. The linear equation of the GCF_1_ group was y = −25,534.553 + 5529.313 × x. The linear equation of the GCF_2_ group was y = −40,522.642 + 6114.495 × x. The linear equation of the GCF_3_ group was y = −19,932.007 + 4389.399 × x.(4)The starting point of the curve was obtained from the linear equation obtained. The difference between the horizontal coordinates of the starting point and the peak point of the curve was the mid-span deflection. The results are shown in Table 7.

The remaining groups used the above method to obtain their respective mid-span deflections. The results are shown in Table 8 and Table 9.

### 3.2. Test Results and Analysis

According to the mid-span deflections (*d*) determined in Section 3.1, the maximum flexural tensile strains (*ɛ*_B_) at the bottom of the beam were calculated using Equation (1) and averaged, as shown in Figure 7. In the winter cold zone, the Chinese specification JTG F40–2004 [42] requires that the destructive strain of modified asphalt mixtures should not be less than 2800 × 10^−6^. The *ɛ*_B_ of all the beam specimens at the time of failure met the specification requirements.

#### 3.2.1. Effect of Geosynthetic Types on ɛ_B_

With the same SCB type conditions, *ɛ*_B_ increases with the increase in tensile strength of the geosynthetics. In the AC-20/AC-25 combination, compared with UN, the *ɛ*_B_ of the FPM, GCF, and CCF increased by 14.94%, 39.95%, and 72.28%, respectively. In the AC-13/AC-20 combination, the *ɛ*_B_ of the FPM, GCF, and CCF increased by 11.81%, 26.51%, and 62.03%, respectively. The above data indicate that the geosynthetic type plays an important role in the improvement of the *ɛ*_B_ of the SCB. The improvement effect of the geogrid is superior to that of the FPM because the mechanism of action between the geogrid and geotextile is different. Geogrid reinforcement is mainly achieved through friction and interlocking, while geotextile reinforcement is mainly achieved through friction [47]. In addition, the tensile strength of the geogrid was higher than that of the FPM. The higher the tensile strength, the stronger the reinforcement effect, and the stronger the stress diffusion effect. Under the same load, the lower the damage to the specimen, the greater the strain generated. The improvement effect of the GCF was lower than that of the CCF, which may be due to differences in the performance of the longitudinal and transverse ribs of the geogrid [6].

#### 3.2.2. Effect of SCB Type on *ɛ*_B_

In the case of the unreinforced SCB, the *ɛ*_B_ of the AC-13/AC-20 combination was slightly lower than that of the AC-20/AC-25 combination, with insignificant differences. In the case of the reinforced SCB, the *ɛ*_B_ of the AC-13/AC-20 combination was lower than that of the AC-20/AC-25 combination, with significant differences, especially in the case of geogrid reinforcement. The main reason is due to the aggregate particle size of the SCB. The coarse aggregate of the AC-13/AC-20 combination was less than that of the AC-20/AC-25 combination. The larger the particle size of the coarse aggregate, the higher the content of the coarse aggregate and the stronger the friction effect, making it more difficult for the aggregate to move [48]. For geogrid reinforcement, the effective particle size range of aggregates varied depending on the mesh size. Research has shown that the ratio of mesh size in bidirectional geogrids to aggregate particle size ranges from 0.96 to 2.0, with a stronger interlocking effect and better reinforcement effect [49,50,51,52]. For the geogrid in this study, the effective particle size range of the aggregate was 13.2–26.5 mm. The aggregate content within the effective particle size range of the AC-13/AC-20 combination was lower than that of the AC-20/AC-25 combination, so the reinforcement effect of the AC-20/AC-25 combination was better and the strain it could withstand was greater.

### 3.3. Two-Way Analysis of Variance

Analysis of variance (ANOVA) is a method of determining whether categorical independent variables have a significant impact on numerical dependent variables by examining whether the means of each population are equal [53]. Therefore, ANOVA was used to determine the extent to which the geosynthetic types and the SCB types affected the low-temperature cracking resistance of asphalt mixtures. The selected indicator was *ɛ*_B_.

Geosynthetic type was selected as the column factor (C), with four levels: UN, FPM, GCF, and CCF. SCB type was chosen as the row factor (R), with two levels: AC-13/AC-20 and AC-20/AC-25. The *ɛ*_B_ was treated as the observation value. Considering that the geosynthetic type and the SCB type paired together have a new effect on *ɛ*_B_, the two-way ANOVA with interactions was used. The data structure is tabulated in Table 10. The hypotheses proposed are as follows:

Geosynthetic type:


**H_0_:**
* The geosynthetic type has no significant influence on ɛ*
_B_



**H_1_:**
* The geosynthetic type has a significant influence on ɛ*
_B_


SCB type:


**H_0_:**
* The SCB type has no significant influence on ɛ*
_B_



**H_1_:**
* The SCB type has a significant influence on ɛ*
_B_


Interaction:


**H_0_:**
* The interaction between the geosynthetic type and the SCB type has no significant influence on ɛ*
_B_



**H_1_:**
* The interaction between the geosynthetic type and the SCB type has a significant influence on ɛ*
_B_


The influences of geosynthetic type and SCB type on *ɛ*_B_ were analyzed using two-way ANOVAs with interactions in Origin software. The results are shown in Table 11.

As can be seen in Table 11, for the geosynthetic type, H_0_ was rejected since F-value > F_critical_-value at the significance level α = 0.05, while the *p*-value tends to be 0 and less than α = 0.05. It is shown that the geosynthetic type has a significant effect on *ɛ*_B_. For the SCB type, H_0_ is rejected as F-value > F_critical_-value at the significance level α = 0.05, while the *p*-value tends to be 0 and less than α = 0.05, indicating that the SCB type had a significant effect on *ɛ*_B_. The interaction reflects the additional effect on *ɛ*_B_ resulting from the combination of the geosynthetic type and SCB type. For the interaction, H_0_ is not rejected at the significance level α = 0.05 due to F-value < F_critical_-value and *p*-value > α = 0.05. It is shown that the interaction between the geosynthetic type and the SCB type does not have a significant impact on *ɛ*_B_.

In addition, the relational strength index *R*^2^ is introduced to reflect the strength of the relationship between the geosynthetic type and the SCB type with *ɛ*_B_, which are calculated as follows:(2)$$R_{C}^{2}=\frac{SSC}{SST}$$(3)$$R_{R}^{2}=\frac{SSR}{SST}$$(4)$$R_{RC}^{2}=\frac{SSRC}{SST}$$(5)$$R_{R+C}^{2}=\frac{SSR+SSC+SSRC}{SST}$$
where *R^2^_C_* is the strength of the relationship between the geosynthetic type and *ɛ*_B_; *R^2^_R_* is the strength of the relationship between the SCB type and *ɛ*_B_; *R^2^_RC_* is the strength of the relationship in the interaction between the geosynthetic type and the SCB type and *ɛ*_B_; *R^2^_R+C_* is the strength of the relationship between the geosynthetic type and the SCB type together and *ɛ*_B_; SST—total sum of square; SSR—sum of squares of the variables for the SCB type; SSC—sum of squares of the variables for the geosynthetic type; and SSRC—sum of squares of the interactions.

The results of the calculations based on Equations (2)–(5) are shown in Table 12. As can be seen from Table 12, *R^2^_R+C_* = 94.08%, indicating that the two factors of geosynthetic type and SCB type together explain a total of 94.08% of the difference in *ɛ*_B_, while the other factors explain only 1.25% of the difference in *ɛ*_B_. Whereas *R_R+C_* = 0.97, which indicates a strong relationship between the geosynthetic type and the SCB type together and *ɛ*_B_. *R_C_* = 0.9424, indicating a strong relationship between geosynthetic type and *ɛ*_B_. *R_R_* = 0.2002, indicating no strong relationship between the SCB type and *ɛ*_B_. Since *R^2^_C_* = 88.82% > *R^2^_R_* = 4.01%, it indicates that the influence of geosynthetic type and SCB type on *ɛ*_B_ can be ranked in descending order as geosynthetic type > SCB type. Therefore, it is recommended that the structural design of geosynthetics reinforced asphalt pavements should focus on the reasonable selection of geosynthetic types to achieve the performance of up to standard and reasonable economic.

## 4. Discussion

Under low-temperature conditions, compared with SCB, RSCB can withstand greater loads and increase the maximum flexural tensile strain, thus improving the low-temperature crack resistance of the SCB. The results of this study are in agreement with those of Nejad [16,18], Zofka [23], Canestrari [25], and Hu [54] et al. Based on statistical methods, the most effective factor affecting the maximum flexural tensile strain in this study is the geosynthetic type, which is in agreement with the findings of Nejad [18]. It is recommended that reasonable geosynthetic types should be selected for the construction of asphalt pavements.

For future studies, it is recommended that more geosynthetic types and SCB types be considered to validate the results of this study. Furthermore, the long-term performance of the carbon fibre geogrid–reinforced asphalt pavements under differential environmental conditions should be taken into account.

## 5. Conclusions

The paper studied the influence of geosynthetic types and SCB types on the low-temperature cracking resistance of reinforced asphalt mixtures based on statistical methods through a series of low-temperature bending damage tests, which helps to improve the durability of carbon fibre geogrid–reinforced asphalt pavement surface structures. The following conclusions can be made accordingly:(1)This study proposes a new method for determining the mid-span deflection, namely the linear fitting difference method.(2)Under the same SCB type conditions, from the perspective of the *ɛ*_B_ index, the interlayer laying of geosynthetics can improve the low-temperature cracking resistance of asphalt pavement SCBs, with the ranked order from highest to lowest being CCF > GCF > FPM > UN.(3)In the case of reinforcement, the low-temperature cracking resistance of AC-20/AC-25 is superior to that of AC-13/AC-20 in terms of the *ɛ*_B_ index, especially in the case of geogrid reinforcement.(4)The two-way ANOVA with interactions shows that the geosynthetic type has a significant impact on the *ɛ*_B_ of asphalt pavement SCBs and that there is a strong relationship between the two. The SCB type has a significant impact on the *ɛ*_B_ of asphalt pavement SCBs, and there is no strong relationship between the two. In addition, the order of influence of the two factors on the *ɛ*_B_ ranks as geosynthetic type > SCB type.

## Figures and Tables

**Figure 1 polymers-17-00461-f001:**
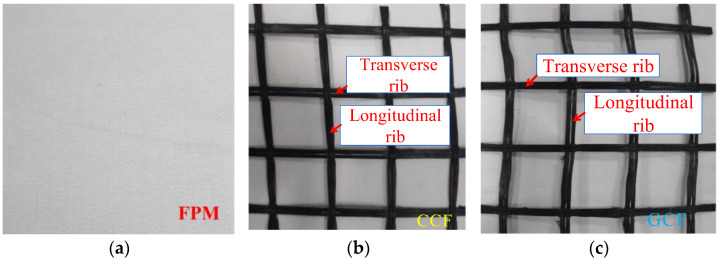
Types of geogrids: (**a**) fibreglass–polyester paving mat; (**b**) carbon fibre geogrid; (**c**) glass/carbon fibre composite qualified geogrid.

**Figure 2 polymers-17-00461-f002:**
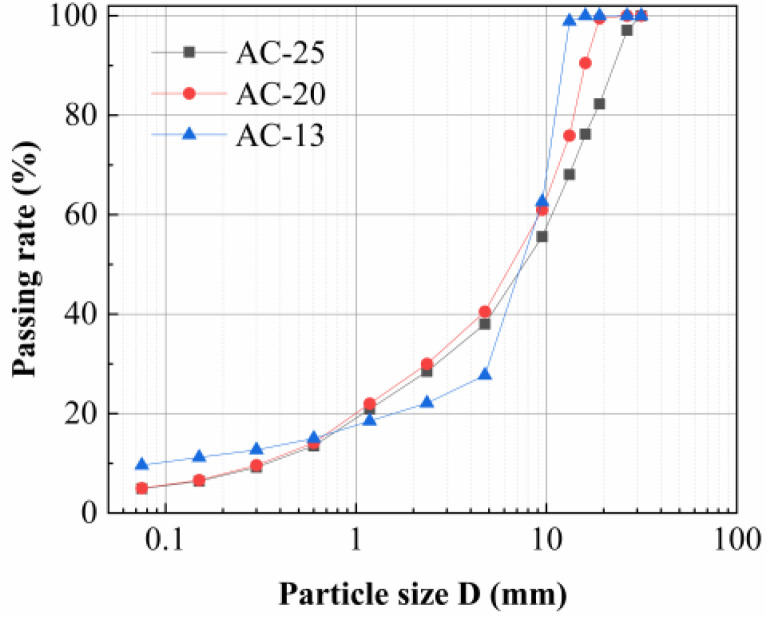
Asphalt mixture aggregate gradation curve diagram.

**Figure 3 polymers-17-00461-f003:**
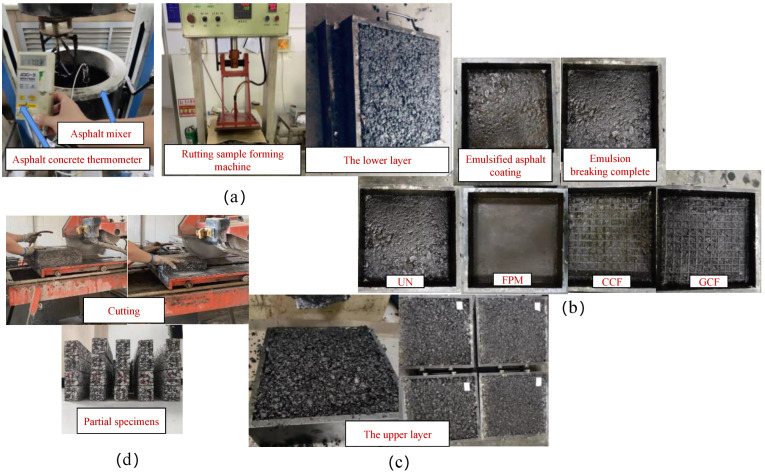
Specimen preparation procedure: (**a**) Mixing and rolling to form the lower layer (**b**) Laying geosynthetics (**c**) Mixing and rolling to form the upper layer (**d**) Plate cutting.

**Figure 4 polymers-17-00461-f004:**
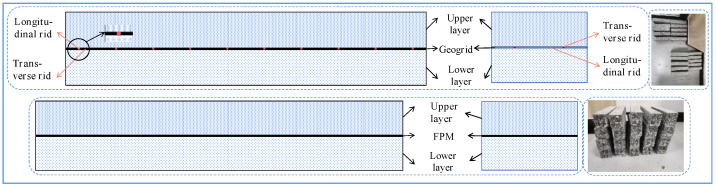
Double-layer reinforced beam specimens.

**Figure 5 polymers-17-00461-f005:**
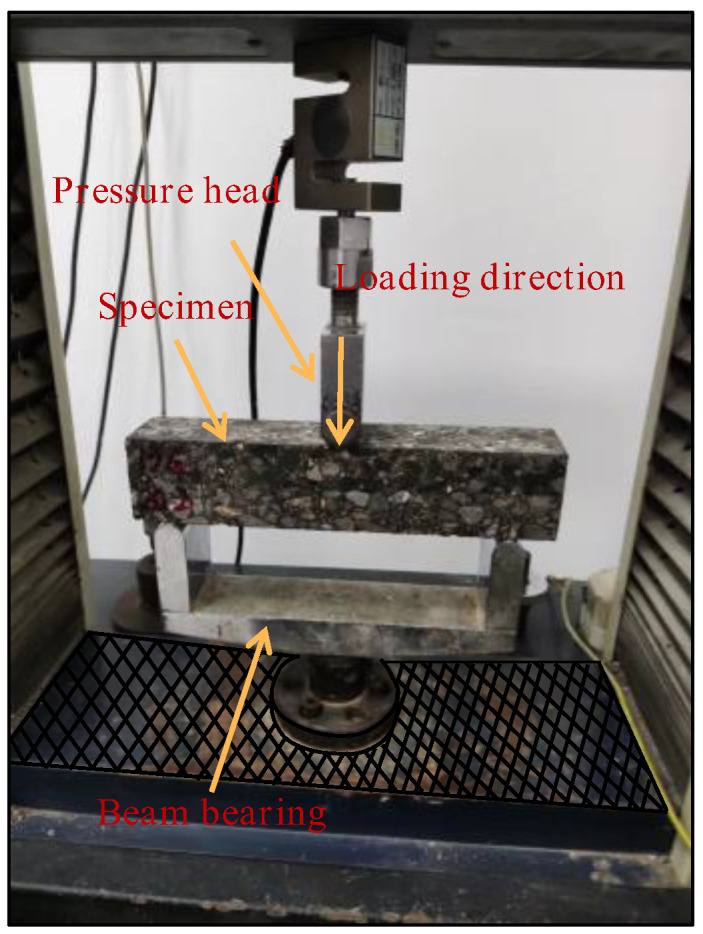
Low-temperature bending damage test [6].

**Figure 6 polymers-17-00461-f006:**
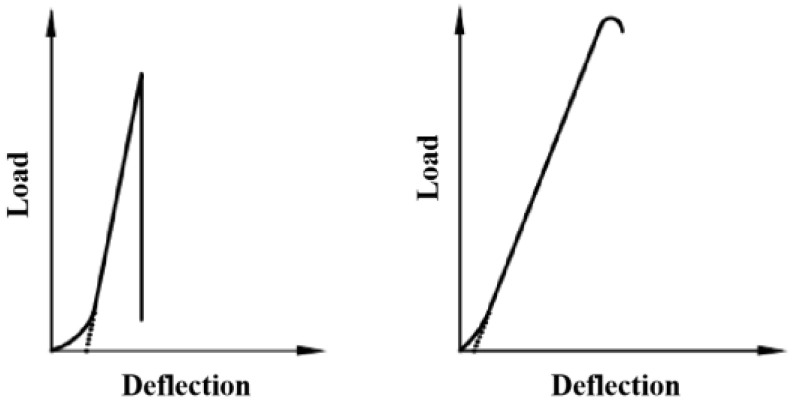
Load–deflection diagram (T0571-1) [41].

**Figure 7 polymers-17-00461-f007:**
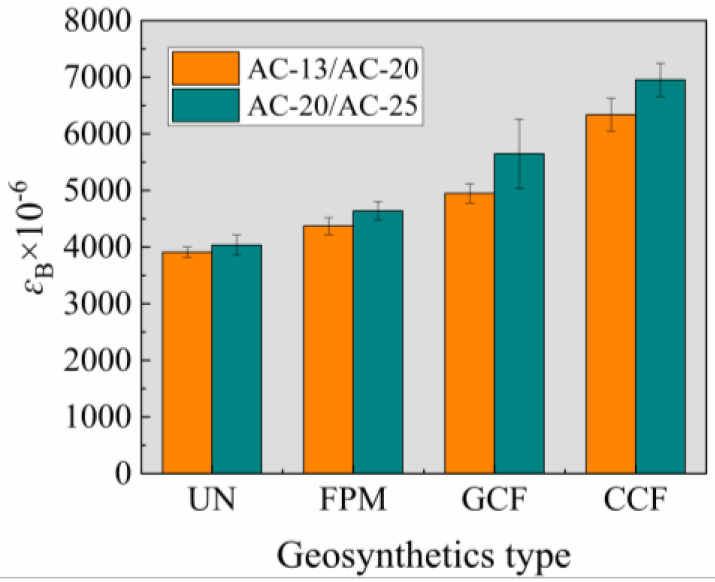
Low−temperature bending damage test results.

**Table 1 polymers-17-00461-t001:** Major characteristics of the fibreglass–polyester paving mat.

Index		FPM
Ultimate tensile strength (kN/m)	Longitudinal	8
	Transverse	8
Ultimate elongation (%)	Longitudinal	≤4
	Transverse	≤4
Mass per unit area (g/m^2^)		25 × 25
Thickness (mm)		1.2

**Table 2 polymers-17-00461-t002:** Major characteristics of geogrids.

Index		GCF	CCF
Ultimate tensile strength (kN/m)	Longitudinal	50	80
	Transverse	80	80
Thickness (mm)		0.6	0.6
Ultimate elongation (%)	Longitudinal	≤3	≤2
	Transverse	≤2	≤2
Aperture size (mm × mm)		25 × 25	25 × 25

**Table 3 polymers-17-00461-t003:** Technical parameters of SBS-modified asphalt.

Items	Measured Value	Standardized Requirement
Softening point (°C)	64.4	≥60
Ductility, 5 cm/min,10 °C (cm)	25.1	≥20
Penetration, 25 °C, 5 s, 100 g (0.1 mm)	57.8	30~60

**Table 4 polymers-17-00461-t004:** Main properties of PCR cationic emulsified asphalt.

Type	Cationic Rapid-Setting Emulsified Asphalt
Content of residual binder (%)	50.1
Sieve test (%)	0.09
Viscosity (Pas)	25
Identification of cationic property	Positive

**Table 5 polymers-17-00461-t005:** Summary of linear fitting results for the GCF reinforcement group of the AC-20/AC-25 combinations (at the 0.05 level).

No. of Groups	Number of Points	Equations	Determination Coefficients	Probability > |t|/ Intercept Distance	Probability > |t|/Slope	Significance
GCF_1_	3	y = −23,757.865 + 5202.387 × x	0.999 71	0.012,77	0.010,93	Yes
	4	y = −24,574.913 + 5354.764 × x	0.999 35	4.33 × 10^−4^	3.25 × 10^−4^	Yes
	5	y = −25,630.259 + 5552.579 × x	0.998 36	4.15 × 10^−5^	2.82 × 10^−5^	Yes
	6	y = −26,822.272 + 5777.036 × x	0.996 75	6.30 × 10^−6^	3.97 × 10^−6^	Yes
	7	y = −27,144.158 + 5837.989 × x	0.997 79	1.33 × 10^−7^	7.78 × 10^−8^	Yes
	8	y = −26,924.466 + 5796.164 × x	0.998 41	2.31 × 10^−9^	1.26 × 10^−9^	Yes
	9	y = −26,392.190 + 5694.304 × x	0.998 12	1.64 × 10^−10^	8.45 × 10^−11^	Yes
	10	y = −25,534.553 + 5529.313 × x	0.996 26	1.11 × 10^−10^	5.37 × 10^−11^	Yes
GCF_2_	3	y = −38,860.375 + 5888.887 × x	0.999 51	0.015,51	0.014,11	Yes
	4	y = −40,013.847 + 6047.389 × x	0.999 34	3.90 × 10^−4^	3.29 × 10^−4^	Yes
	5	y = −41,044.743 + 6189.585 × x	0.999 14	1.35 × 10^−5^	1.07 × 10^−5^	Yes
	6	y = −41,964.464 + 6316.914 × x	0.998 94	5.55 × 10^−7^	4.19 × 10^−7^	Yes
	7	y = −42,693.542 + 6418.189 × x	0.998 86	2.06 × 10^−8^	1.50 × 10^−8^	Yes
	8	y = −42,170.353+ 6345.220 × x	0.998 93	5.52 × 10^−10^	3.87 × 10^−10^	Yes
	9	y = −40,522.642 + 6114.495 × x	0.995 88	1.94 × 10^−9^	1.31 × 10^−9^	Yes
	10	y = −37,857.247 + 5739.828 × x	0.985 65	1.81 × 10^−8^	1.16 × 10^−8^	Yes
GCF_3_	3	y = −20,211.300+ 4444.125 × x	0.989 32	0.077,14	0.065,91	Yes
	4	y = −20,636.599 + 4523.500 × x	0.995 52	0.003,01	0.002,24	Yes
	5	y = −20,850.990 + 4563.715 × x	0.997 67	7.22 × 10^−5^	4.76 × 10^−5^	Yes
	6	y = −20,866.564 + 4566.651 × x	0.998 67	1.13 × 10^−6^	6.67 × 10^−7^	Yes
	7	y = −19,932.007 + 4389.399 × x	0.996 77	3.85 × 10^−7^	2.02 × 10^−7^	Yes
	8	y = −19,532.960 + 4313.399 × x	0.997 1	1.60 × 10^−8^	7.60 × 10^−9^	Yes
	9	y = −18,795.426 + 4172.176 × x	0.995 14	5.48 × 10^−9^	2.34 × 10^−9^	Yes
	10	y = −17,809.800+ 3982.429 × x	0.990 16	6.72 × 10^−9^	2.57 × 10^−9^	Yes

**Table 6 polymers-17-00461-t006:** Summary of the calculation results of the linear equation of the GCF reinforcement group for the AC-20/AC-25 combination.

No. of Groups	Number of Points	Predicted Value	Difference	Percentage
GCF_1_	3	4428.667	4.333	0.10%
	4	4437.200	12.866	0.29%
	5	4453.614	29.280	0.66%
	6	4477.709	53.375	1.21%
	7	4486.069	61.735	1.40%
	8	4479.152	54.818	1.24%
	9	4459.550	35.216	0.80%
	10	4423.265	1.069	0.02%
GCF_2_	3	4322.833	6.333	0.15%
	4	4331.656	15.156	0.35%
	5	4343.485	26.985	0.63%
	6	4357.465	40.965	0.95%
	7	4371.040	54.540	1.26%
	8	4359.147	42.647	0.99%
	9	4314.953	1.547	0.04%
	10	4232.911	83.589	1.94%
GCF_3_	3	3862.528	17.861	0.46%
	4	3867.199	22.532	0.59%
	5	3870.653	25.986	0.68%
	6	3870.986	26.319	0.68%
	7	3845.366	0.699	0.02%
	8	3832.723	11.944	0.31%
	9	3805.253	39.414	1.03%
	10	3763.018	81.649	2.12%

**Table 7 polymers-17-00461-t007:** Summary of deflections of the GCF group for the AC-20/AC-25 combination.

No. of Groups	Equations	Starting Point	Peak Point	Deflection/mm
GCF_1_	y = −25,534.553+ 5529.313 × x	4.618	5.418	0.800
GCF_2_	y = −40,522.642+ 6114.495 × x	6.627	7.333	0.706
GCF_3_	y = −19,932.007 + 4389.399 × x	4.541	5.417	0.876

**Table 8 polymers-17-00461-t008:** AC-13/AC-20 combination deflections.

No. of Groups	Geosynthetic Type			
	UN	FPM	GCF	CCF
1	0.508	0.592	0.692	0.897
2	0.531	0.563	0.659	0.853
3	0.532	0.561	0.633	0.853
4	0.526	0.594	0.668	0.791
5	0.510	0.605	0.646	0.830

**Table 9 polymers-17-00461-t009:** AC-20/AC-25 combination deflections.

No. of Groups	Geosynthetic Type			
	UN	FPM	GCF	CCF
1	0.518	0.621	0.800	0.971
2	0.530	0.610	0.706	0.926
3	0.573	0.652	0.876	0.878
4	0.551	0.590	0.697	0.961
5	0.519	0.620	0.687	0.900

**Table 10 polymers-17-00461-t010:** Data structure table of the two-way ANOVA with interactions.

SCB Type	Geosynthetic Type			
	UN	FPM	GCF	CCF
AC-13/AC-20	3810	4440	5190	6727.5
	3982.5	4222.5	4942.5	6397.5
	3990	4207.5	4747.5	6397.5
	3945	4455	5010	5932.5
	3825	4537.5	4845	6225
AC-20/AC-25	3885	4657.5	5152.5	7282.5
	3975	4575	6000	6945
	4297.5	4890	5295	6585
	4132.5	4425	5227.5	7207.5
	3892.5	4650	6570	6750

**Table 11 polymers-17-00461-t011:** Two-way analyses of variance with interactions.

Sources of Error	Degrees of Freedom/*df*	Square Sum/*SS*	Mean Square/*MS*	F-Value	*p*-Value	F_critical_-Value
Geosynthetic type (*C*)	3	4.05 × 10^7^	1.35 × 10^7^	161.31	0	2.845
SCB type (*R*)	1	1.83 × 10^6^	1.83 × 10^6^	21.92	4.99 × 10^−5^	4.091
Interaction (*RC*)	3	5.71 × 10^5^	1.90 × 10^5^	2.27	0.10	2.845
Error (*E*)	32	2.68 × 10^6^	83,648.67	--	--	--
Summation (*T*)	39	4.56 × 10^7^	--	--	--	--

**Table 12 polymers-17-00461-t012:** Results of the relationship strength index *R*^2^.

*R*^2^ Type	*R* ^2^ * _C_ *	*R* ^2^ * _R_ *	*R* ^2^ * _RC_ *	*R* ^2^ * _R+C_ *
Calculated value (%)	88.82	4.01	1.25	94.08

## Data Availability

The original contributions presented in this study are included in the article. Further inquiries can be directed to the corresponding author.

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
