# Peer review of "Study on Low-Temperature Cracking Resistance of Carbon Fibre Geogrid Reinforced Asphalt Mixtures Based on Statistical Methods"

_polymers, 2025, doi:10.3390/polym17040461_

Round 1

Reviewer 1 Report

Comments and Suggestions for Authors

The study investigates the low-temperature cracking resistance of asphalt mixtures with carbon fiber geogrids. The research aspect is indeed essential in asphalt pavement engineering, but a lot of areas still require further elaboration and critical assessment of the work:

* The problem of cracking in asphalt pavements is well diagnosed concerning its significance for the life of the pavement and the structural integrity. The manuscript would, however, benefit from a more extensive review of the literature, incorporating a wider spectrum of studies on the factors that influence cracking: types of asphalt, composition, and aging effects. The references currently cited relate predominantly to the role of the geogrid, which is significant on its own terms yet does not give a full perspective on the problem.

* This section on methodology is vague when it states pertaining to an experimental design and specific measured parameters. For example, the mention of low-temperature bending damage tests is not specified under which condition these tests were carried out or quantified. Amounting to a much deeper explanation of the testing protocols and reasons behind certain geosynthetics would make the study more replicable.

* The results showing that adding geosynthetics, especially carbon fiber geogrids, can improve cracking resistance, could be discussed further from the perspective of these results against those in already-published literature that deals with performance in similar applications using different geosynthetic materials, such as glass geogrids. Such a comparison provides a better context for the importance of the present results.

* The manuscript points out the economy of using textile-grade polyacrylonitrile (PAN) fibers as precursors for carbon fibers. However, there is no discussion of the economic feasibility of incorporating these materials into real-life applications. A cost-benefit analysis would be an excellent addition to the sophisticated take on academia that would help practitioners in the field and, more importantly, help in making the findings of this research more practicable.

* Nonetheless, the final view reiterates the most salient points concerning the findings, while falling short of any sufficient elaboration on the implications of such findings on future research and ultimately practice in pavement engineering. Further directions of study would include suggestions of additional research exploring the long-term performance of such materials under differential environmental conditions.

Author Response

Please see the attachment, thank you!

Reviewer 2 Report

Comments and Suggestions for Authors

The study explores the influence of carbon fiber geogrid on low-temperature cracking resistance, which is relatively underexplored compared to other geosynthetics.

The use of low-temperature bending damage tests combined with statistical methods like two-way ANOVA provides robust data analysis.

The findings detail the comparative performance of different geosynthetics, including fiberglass-polyester paving mat (FPM), glass/carbon fiber composite geogrid (GCF), and carbon fiber geogrid (CCF).

The study goals are well-articulated, aiming to establish a theoretical basis for selecting reinforced asphalt pavement structures.

Figures, tables, and mathematical derivations enhance the clarity of experimental results.

While the introduction highlights the significance of the study, it could better integrate recent advancements and global perspectives on carbon fiber geogrid applications.

The preparation of double-layer reinforced beam specimens lacks sufficient visual or procedural details.

Although the statistical analysis is comprehensive, the implications of the findings for practical applications could be expanded.

The two-way ANOVA suggests no significant interaction between geosynthetics and SCB types, but the discussion on this finding implications is minimal.

Some typographical errors and inconsistent formatting (e.g., figure captions, equation numbering).

The manuscript would benefit from professional language editing to improve fluency and technical clarity.

Suggestions for improvement:

Include more recent studies on carbon fiber geogrid applications and their global relevance to strengthen the introduction and discussion (see, e.g., [1,2]).

[1] Giorgio, I., Ciallella, A. & Scerrato, D. (2020) A study about the impact of the topological arrangement of fibers on fiber-reinforced composites: Some guidelines aiming at the development of new ultra-stiff and ultra-soft metamaterials, International Journal of Solids and Structures, 203:73–83.

[2] dell'Isola, F., Steigmann, D., & Della Corte, A. (2015). Synthesis of fibrous complex structures: designing microstructure to deliver targeted macroscale response. Applied Mechanics Reviews, 67(6), 060804.

Provide a step-by-step visual or schematic representation of the specimen preparation and testing processes.

Discuss how the findings could influence pavement design practices, especially in regions prone to low temperatures.

Ensure consistent formatting of figures, tables, and equations for better readability.

Elaborate on the practical significance of the ANOVA results, especially regarding non-significant interactions.

The paper contributes significantly to understanding the low-temperature cracking resistance of carbon fiber geogrid-reinforced asphalt mixtures. However, it requires improvements in contextualization, clarity of methods, and practical implications to maximize its impact. With these revisions, it could become a valuable resource for researchers and engineers in pavement materials science.

Comments on the Quality of English Language

The manuscript would benefit from professional language editing to improve fluency and technical clarity.

Author Response

Please see the attachment, thank you!

Round 2

Reviewer 1 Report

Comments and Suggestions for Authors

The authors have made efforts to improve the article and eliminate its deficiencies. The article is acceptable.